# Congenital Diaphragmatic Hernia: Perinatal Prognostic Factors and Short-Term Outcomes in a Single-Center Series

**DOI:** 10.3390/children10020315

**Published:** 2023-02-07

**Authors:** Camilla Pagliara, Elisa Zambaiti, Giulia Brooks, Luca Bonadies, Costanza Tognon, Sabrina Salvadori, Alberto Sgrò, Francesco Fascetti Leon

**Affiliations:** 1Department of Pediatric Surgery, Universitary Hospital of Padova, 35128 Padova, Italy; 2Department of Pediatric Surgery, Paediatric Hospital of Torino, 10126 Torino, Italy; 3Department of Neonatal Intensive Care, Universitary Hospital of Padova, 35128 Padova, Italy; 4Department of Pediatric Anesthesiology, Universitary Hospital of Padova, 35128 Padova, Italy

**Keywords:** congenital diaphragmatic hernia, diaphragmatic patch, dopamine, inotropic agents, milrinone, pulmonary hypertension

## Abstract

**Background**: Many prognostic factors for CDH patients are described and validated in the current literature: the size of diaphragmatic defects, need for patch repair, pulmonary hypertension and left ventricular dysfunction are recognized as the most influencing outcomes. The aim of this study is to analyze the influence of these parameters in the outcome of CDH patients in our department and identify any further prognostic factors. **Methods**: An observational retrospective single-center study was conducted including all patients treated at our centre with posterolateral CDH between 01.01.1997 and 12.31.2019. The main outcomes evaluated were mortality and length of hospital stay. A univariate and multivariate analysis was performed. **Results**: We identified 140 patients with posterolateral CDH; 34.8% died before discharge. The overall median length of stay was 24 days. A univariate analysis confirmed that both outcomes are associated with the size of diaphragmatic defects, need for patch repair and presence of spleen-up (*p* < 0.05). A multivariate analysis identified that the need for patch repair and maximum dopamine dose used for cardiac dysfunction are independent parameters associated with the length of stay only (*p* < 0.001). **Conclusions**: In our series, the duration of hospitalization is longer for newborns with CDH treated with higher doses of dopamine for left ventricular dysfunction or needing patch repair in large diaphragmatic defects.

## 1. Introduction

Congenital diaphragmatic hernia (CDH) is a rare congenital malformation characterized by a defective diaphragmatic development, with a total prevalence of about one case in every 2500–4000 births [1]. The reported mortality is between 20 and 60%, mostly depending on the hospital volume [2,3]. Consequently, in recent decades, almost all centers have adopted standardized protocols developed by the Medical Group (i.e., CDH Study Group, ERNICA Group, CDH Euro Consortium) for prenatal and neonatal management and short- and long-term follow-up. This has contributed to reducing the gap between different centers in terms of mortality and morbidity [4,5].

Nevertheless, numerous causes still contribute to the poor outcome of this condition. Several studies have tried to identify the prognostic factors of short-term outcomes in neonates with CDH. Among those, the two main factors that heavily influence the prognosis of CDH patients are pulmonary hypoplasia and pulmonary hypertension (PH); the latter is mainly due to vessel structural changes associated with parenchymal hypoplasia [6,7]. To a lesser extent, some intraoperative findings, such as localization of the liver and spleen, size of diaphragmatic defects and need for patch repair, have been validated as predictive factors of mortality and morbidity among this group of patients [8,9]. Finally, due to the innovative technology related to prenatal diagnosis developed in recent years, some prenatal indices, such as observed-to-expected lung-to-head ratio (O/E LHR) and total fetal lung volume (TFLV), have been widely accepted as prognostic factors [10,11].

The presence of herniated abdominal content in the thorax is known to affect not only lung development but also the cardiac one: newborns with CDH often present with a left ventricular hypoplasia with a reduced systolic performance [12]; according to this, the employment of inotropic agents (i.e., dopamine, dobutamine, noradrenaline, milrinone) is common in these neonates in order to support cardiac function and prevent hypotensive crisis [13].

The aim of our study is to analyze the clinical characteristics, intraoperative findings and therapeutic choices in patients treated at our department according to our protocol in order to define which parameter could influence mortality and the length of stay among our cohort of patients and identify any further prognostic factors not validated in the literature.

## 2. Materials and Methods

An observational retrospective cohort study was conducted on all patients treated in the neonatal period for posterolateral CDH between 1 January 1997 and 31 December 2019. Analysis was based on data from a single tertiary centre. We excluded all patients undergoing repair beyond 30 days of age, presenting with Morgagni hernias, true diaphragmatic eventratio and diaphragmatic relaxatio.

To achieve cardiopulmonary stability before performing surgical repair, a standardized protocol was employed for all in-born neonates, consisting of immediate endotracheal intubation at birth and mechanical ventilation. Neonates unresponsive to conventional ventilation were submitted to high-frequency oscillatory ventilation (HFOV). Extracorporeal membrane oxygenation (ECMO) was adopted in case of persistent cardiopulmonary instability, despite HFOV. For out-born patients, most of them were reported at birth and were treated with a similar protocol after being transferred to our center.

We collected variables in three main periods: pre, intra and post-operatively. Pre-operative data were O/E LHR, gestational age, weight at birth, sex, presence of associated malformations, probability of survival score (POS), ECMO requirement at birth and presence of PH. O/E LHR was calculated according to the formula proposed by Kehl et al. [14]. Following the equation proposed by the CDH study group, POS was calculated with APGAR values at 1 and 5 min [15]. In patients intubated and ventilated at birth, APGAR values were calculated according to the guidelines of American Academy of Pediatrics [16]. The presence and severity of PH was determined using postnatal echocardiography, performed within the first 24 h, and need for treatment using pulmonary vasodilators (sildenafil or inhalator nitric oxide—iNO_2_) (Table 1).

As intra-operative parameters, we evaluated CDH side, size of diaphragmatic defects, localization of liver and spleen at surgery (liver-up/spleen-up), presence of hernia sac, need for patch during surgery and length of surgical procedure. Size of diaphragmatic defect was rated according to CDH Study Group classification [17]. Post-operatively, we considered the total length of mechanical ventilation, the need for oxygen therapy at 30 days and the use and dose of inotropes.

Main outcomes were mortality and length of hospital stay. The binary outcome of length of stay (more or less than 3 weeks) was defined arbitrarily, using a value similar to the median length of stay of our patients (24 days, IQR: 17–33).

Categorical variables were reported as proportion, while continuous variables as medians with their range. Outcomes between groups were compared using Χ² test, Fischer’s exact test and Mann–Whitney U test as appropriate.

Multivariable logistic regression was performed for the outcomes of death and length of hospital stay. The covariates examined in the regression model were size of diaphragmatic defects, need for patch repair, severity of PH, use of iNO_2_, need for O_2_ therapy at 30 days and dopamine maximum dose.

Statistical analysis was conducted using GraphPad Prism 8.3.0. *p*-value < 0.05 was considered statistically significative. The study was conducted in accordance with the 1964 Helsinki Declaration and local regulations. Due to the retrospective design of the study, the project was notified to the ethics board of both centers and required no formal ethical approval. Informed consensus was obtained for all participants included in the study.

## 3. Results

In the study period, 170 neonates were treated for CDH at our department. One hundred and forty patients suited inclusion criteria, while thirty of them (17.6%) were excluded from the study: five patients presented with a Morgagni (antero-lateral CDH); seven neonates were diagnosed with diaphragmatic eventratio/relaxatio; and eighteen patients were diagnosed with CDH after 30 days from birth.

In total, 49 of 140 patients (35%) died before discharge, 34 of them even before corrective surgery (24.2% of the total) (Figure 1.) Eight patients (5.6%) needed an ECMO at some point during hospitalization, and of these, only two survived until discharge.

### 3.1. Demographics

Ninety neonates (64.2%) had a prenatal diagnosis of CDH. There were 85 males (60.7%), with a female-to-male ratio of 1:2. The median gestational age at birth was 38 weeks (IQR 36.5–39), and the median birth weight was 2900 g (IQR 2428–3265). Thirty-three babies (23.5%) had other associated malformations, such as tetralogy of Fallot (2; 1.4%), interventricular defects (3; 2.2%) and trisomy 18 (3, 2.2%). The median APGAR at 1 and 5 min was, respectively, 6 and 8 (IQR 4–10). The median POS was about 75% (IQR 51–85).

One hundred and seventeen neonates (83.5%) presented with a left CDH, which constitutes a six-times increased incidence than the right ones. Among the other 23, 2 cases of bilateral hernia were identified. A hernia sac was described in 18 cases (12.8%).

### 3.2. Intraoperatives

One hundred and six patients (75.7%) underwent corrective surgery. Forty of them needed for a patch repair (37.7%). The liver and spleen were localized in the thorax, respectively, in 64 (45.7%) and 86 (61.4%) cases either during surgery or autopsy. In eighty-one cases (57.9%), the defect was classified as major, namely, type C/D. After surgery, the median duration of mechanical ventilation was 9 days (IQR 6–14).

### 3.3. Postoperatives

Seventy-three patients (53.5%) were diagnosed with PH. In 41 (56% of the diagnosed cases), the PH was severe and needed treatment with inhalator nitric oxide. Nine children still required supplemental oxygen after 30 days of life, and two of them died during the stay. Inotropes (dopamine and dobutamine) were used in 110 neonates to support circulation. The median of dopamine and dobutamine maximum doses were, respectively, 10 γ/kg/min (IQR 1–14) and 6 γ/kg/min (IQR 5–9). The median length of stay was 24 days (IQR 21–33).

### 3.4. Predictors of Outcomes

The relationship between the two outcomes, mortality and length of stay, and total population features is reported in Table 2, Table 3 and Table 4 for the first outcome, and Table 5, Table 6, and Table 7 for the latter. 

The size of diaphragmatic defects, need for patch repair during surgery, presence and degree of PH, treatment with iNO_2_ and inotropic agents and dopamine maximum doses were significantly associated with both the primary and secondary outcomes.

### 3.5. Multivariable Logistic Regression

We also conducted a multivariable logistic regression for both outcomes: mortality and length of hospital stay. While the analysis did not identify any significative results for the first outcome (mortality), two of our prognostic factors of the univariate analysis were confirmed as independent factors associated with length of stay: need for patch repair (OR 14.7–IC 95% 3.78–98.34) and dopamine maximum dose (OR 0.88–IC 95% 0.78–0.97).

## 4. Discussion

The present study identified the size of diaphragmatic defects and the development of PH and left ventricular dysfunction, characterized by the need for higher doses of inotropic drugs in the neonatal intensive care unit, as strongly influencing mortality and the length of hospitalization among patients with CDH.

Despite the real mortality of patients with CDH being unknown as many epidemiological studies do not include all fetuses who died before delivery or during the voluntary interruption of pregnancy, the reported post-natal mortality in third-level centers is less than 40%, which is similar to the one we documented; the demographic of our population is also comparable to that available in the literature [1,18].

Size defects and the need for patch repair are intimately connected. Synthetic material patches are frequently employed to treat larger defects where primary closure is not achievable due to a scarce diaphragmatic muscular rim [19]. In our series, neonates with a larger diaphragmatic defect (defined as type C and D) and need for patch repair during surgery had a shorter recovery and higher mortality rate compared to those who underwent primary closure. Furthermore, the multivariate analysis confirmed that the association between the need for patch repair and length of stay is independent from the other predictors. This result agrees with what was reported previously by Bradley et al.: in their paper, the author also identified the need for patch repair as an independent predictor of mortality and morbidity in terms of length of mechanical ventilation, need for O2 therapy at 30 days and length of hospitalization [9].

Many studies have identified liver herniation as one of the main indicators linked to a poor outcome. Some authors have hypothesized that the prognostic role of liver herniation may be related both to anatomical changes in the organ itself and an active ingrowth of the liver into the thorax, occurring when diaphragmatic contact misses [20]. Moreover, it seems that liver herniation induces a distortion of fetal vasculature and hemodynamic changes that influences post-operative management [20,21]. In our study, localization of liver was confirmed to be one of the predictors of mortality in infants with CDH.

There are only a few studies that have discussed the impact of the presence of the hernial sac on the prognosis of CDH patients. A sac has been reported to be present in about 20% of infants with CDH [22], while in our study, the prevalence was as low as 12.8%. Recently, some authors have demonstrated that patients with CDH and hernia sacs have a better prognosis, defined as a reduced rate of mortality, less need for patch repair and mechanical ventilation after surgery and shorter hospitalization [23,24]. Probably due to the low prevalence of hernia sacs in our population, our results do not reach statistical relevance, but it seems that newborns with hernia sacs tend to present a lower mortality rate.

Modern advances in neonatal intensive care and surgical techniques have led to great improvement in terms of survival and short-term outcomes among children with CDH, even if the prognosis is still heavily influenced by the presence of pulmonary hypoplasia and the severity of PH [25]. PH in CDH patients is sustained by anatomic and histological anomalies that compromise lung development, both ipsilateral and contralateral: a decrease in bronchiolar generation and a decrease in pulmonary arteriole number and size. The hypertrophy of the pulmonary artery, muscularization of small arterioles and adventitial wall thickening are well-described in both animal and human models [7,26]. A recent study by the CDH study group evaluated patients with “low risk” CDH (defined as defect size A and B) and determined that the prevalence of severe PH on the first postnatal echocardiography was significantly higher in the group of patients who died during the stay or were hospitalized for more than eight weeks [27]. We found a similar relationship in our center, where the presence and severity degree of PH were correlated significantly to both outcomes, defined as death and hospitalization for longer than 3 weeks. According to this result, PH treatment (iNO and sildenafil) was significantly lower among patients discharged within 3 weeks, and none of them needed an oxygen supply at home.

Besides PH, left ventricular hypoplasia and biventricular dysfunction are present in most patients with severe CDH [12]. In order to prevent hypotensive crisis and sustain cardiac performance, inotropic agents have been frequently used, with a considerable preference for dopamine [13,28]. The pharmacodynamic effects of dopamine are dose-dependent: for doses <3 γ/kg/min, it induces renal and cerebral vasodilatation, while for doses >5 γ/kg/min, it increases myocardial contractility and determines peripheral vasoconstriction [29]. The median dopamine maximum dose used in NICU was about 10 γ/kg/min; significantly higher doses were used for patients hospitalized for more than 3 weeks. Moreover, dopamine maximum dose (γ/kg/min), employed to sustain peripheral circulation, has been identified by the multivariate logistic regression analysis as an independent predictor of length of stay. The recent studies suggest that dopamine increases peripheral resistances in systemic circulation, but it also has collateral effects on microcirculation and pulmonary hypertension. This can potentially lead to a worse ventricular performance [30]. According to this new evidence, in recent years, milrinone, in association with noradrenaline, has been used in our center as inotropic agents for neonates with CDH. Milrinone is a phosphodiesterase 3 inhibitor; in addition to its inotropic effect on cardiac muscle, it functions as a pulmonary and systemic dilatator. As an inotropic agent, it acts directly to improve left ventricular kinesis, while its consequences on pulmonary circulation leads to the enhancement of the right ventricle diastolic function, reducing the afterload [31].

Among the limits of the present study, its retrospective and single-center nature does not allow a definitive conclusion to be drawn. In particular, the series includes patients born in over a 20 year period, and some data, mostly those concerning the oldest patients, were missing or incomplete. Furthermore, during this time span, the internal protocol for the management of newborns with CDH has been changed more than once, introducing innovation in terms of mechanical ventilation and therapies for PH. Moreover, the O/E LHR and prenatal liver herniation are universally recognized as prognostic indices, and they are crucial during prenatal counselling in order to comment with parents the prognosis of babies with CDH. Even if about 60% of our patients presented a prenatal diagnosis of a diaphragmatic defect at birth, we were able to recover the O/E LHR value for only half of them. The lack of prenatal data on the majority of the series made it impossible for us to draw conclusions about their prognostic value; thus, they have not been considered in our analysis.

Notwithstanding the above, our results could be useful in postnatal counselling in order to discuss the prognosis of these patients in terms of survival and length of hospital stay. Moreover, our results about inotropic medications in these infants may confirm the need for a deeper understanding of the cardiac involvement in CDH with the aim of identifying the most appropriate approach to treat left ventricular dysfunction in infants who develop severe PH.

## 5. Conclusions

The present study confirms that the size of diaphragmatic defects and the presence of PH and left ventricular dysfunction, characterized by the need for higher doses of inotropic drugs, strongly influence mortality and length of hospitalization among patients with CHD. In particular, in the multivariate analysis, it is evident how the length of stay is longer for patients with CDH treated with higher doses of dopamine for left ventricular dysfunction or needing patch repair for large diaphragmatic defects.

## Figures and Tables

**Figure 1 children-10-00315-f001:**
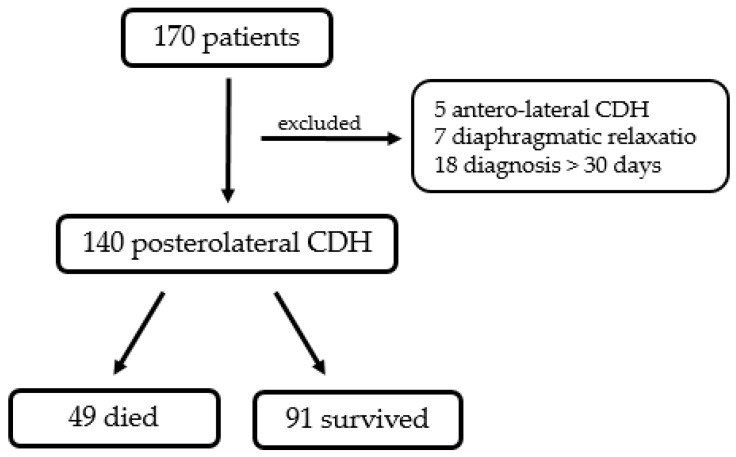
Diagram representing population of the current study.

**Table 1 children-10-00315-t001:** Classification of pulmonary hypertension.

Pulmonary Hypertension	Definition
Low Pulmonary Hypertension	Tricuspid insufficiency, right-to-left shunt through patent arteriosus duct
Mild Pulmonary Hypertension	Tricuspid insufficiency, right-to-left shunt through patent arteriosus duct, right-to-left shunt through patent foramen ovale. NO < 20 ppm
Severe Pulmonary Hypertension	Tricuspid insufficiency, right-to-left shunt through patent arteriosus duct, right-to-left shunt through patent foramen ovale. NO > 20 ppm, sildenafil.

NO = nitric oxide; ppm = part per million.

**Table 2 children-10-00315-t002:** Demographics related to mortality.

	Overall (140)	Dead (49)	Survivor (91)	*p* Value
Prenatal Diagnosis, n (%)	90 (64.2%)	42 (85.7%)	48 (52.7%)	**<0.001 ***
O/E LHR, %, median (IQR)	33 (27–47)	30 (25–34)	44 (30–49.5)	**0.018 #**
Associated Anomalies, n (%)	33 (23.5%)	21 (42.8%)	12 (13.1%)	**<0.001 ***
GA at Birth, weeks, median (IQR)	38 (36.5–39)	37 (35.5–39)	38 (37–39)	**0.002 #**
Weight, grams, median (IQR)	2900 (2428–3265)	2620 (2135–3105)	3000 (2655–3358)	**0.023 #**
Male, n (%)	85 (60.7%)	30 (61.2%)	55 (60%)	0.927 *
POS, %, median (IQR)	75 (51–85)	57 (34–73.25)	80 (68.25–86)	**<0.001 #**
ECMO, n (%)	8 (5.7%)	6 (12.2%)	2 (2.1%)	0.22 °

n = number, O/E LHR = observed-to-expected lung-to-head ratio, GA = gestational age, IQR = interquartile range, POS = probability of survival, ECMO = extracorporeal membrane oxygenation. * X ^2^ test, # Mann–Whitney test, ° Fisher’s exact test.

**Table 3 children-10-00315-t003:** Intraoperatives related to mortality.

	Overall (140)	Dead (49)	Survivor (91)	*p* Value
Side				0.095 °
Left, n (%)	117 (83.6%)	38 (77.6%)	79 (86.8%)	
Right, n (%)	21 (15%)	9 (18.3%)	12 (13.2%)	
Bilateral, n (%)	2 (1.4%)	2 (4.1%)	0 (0%)	
Hernia Sac, n (%)	18 (12.8%)	3 (6.1%)	15 (16.4%)	*0.08 **
Size				**<0.001 ***
A, n (%)	17 (12.1%)	2 (4.1%)	15 (16.5%)	
B, n (%)	42 (30%)	4 (8.2%)	38 (41.7%)	
C, n (%)	52 (37.2%)	24 (48.9%)	28 (30.8%)	
D, n (%)	29 (20.7%)	19 (38.8%)	10 (11%)	
Liver-up, n (%)	64 (45.7%)	34 (69.3%)	30 (32.9%)	**<0.001 ***
Spleen-up, n (%)	86 (61.4%)	29 (59.1%)	57 (62.6%)	0.97 *
Patch, n (%)	40 (37.7%)	11 (73.3%)	29 (31.8%)	**0.002 ***

n = number, * X ^2^ test, ° Fisher’s exact test.

**Table 4 children-10-00315-t004:** Postoperatives and therapeuticals related to mortality.

	Overall (140)	Death (49)	Survival (91)	*p* Value
PH				**<0.001 °**
None, n (%)	64 (45.7%)	1 (2%)	63 (69.2%)	
Mild, n (%)	10 (7.1%)	0 (0%)	10 (11%)	
Moderate, n (%)	21 (15%)	14 (28.6%)	7 (7.7%)	
Severe, n (%)	44 (31.4%)	34 (69.4%)	10 (11%)	
Unknown, n (%)	1 (0.8%)	0 (0%)	1 (1.1%)	
iNO_2_, n (%)	62 (44.2%)	46 (93.8%)	16 (10.9%)	**<0.001 ***
Sildenafil, n (%)	14 (10%)	11 (22.4%)	3 (3.2%)	**<0.001 °**
Inotropes, n (%)	110 (78.5%)	46 (93.8%)	64 (70.3%)	**0.002 ***
Dopamine, γ/kg/min, median (IQR)	10 (1–14)	14 (10–14.5)	7 (0–10)	**<0.001 #**

N = number, PH = pulmonary hypertension, iNO_2_ = inhalator nitric oxide, IQR = interquartile range, * X ^2^ test, # Mann–Whitney test, ° Fisher’s exact test.

**Table 5 children-10-00315-t005:** Demographics related to length of stay.

	Overall (91)	>3 Weeks (54)	≤3 Weeks (37)	*p* Value
Prenatal Diagnosis, n (%)	48 (52.7%)	33 (61.1%)	15 (40.5%)	0.53 *
O/E LHR, %, median (IQR)	26 (10–45)	45 (30–49.5)	34.5 (27–58.5)	0.67 #
Associated Anomalies, n (%)	12 (13.1%)	10 (18.5%)	2 (5.4%)	0.11 °
GA at Birth, weeks, median (IQR)	38 (36–39)	38 (37–39)	38 (37–39)	0.25 #
Weight, grams, median (IQR)	3000 (2620–3470)	2980 (2520–3270)	3100 (2885–3400)	0.23 #
Male, n (%)	55 (60%)	32 (59.2%)	23 (62.1%)	0.78 *
POS, %, median (IQR)	80 (66–89)	74 (59–84)	85 (78–90)	0.12 #
ECMO, n (%)	2 (2.1%)	2 (3.7%)	0 (0%)	0.51 °

n = number, O/E LHR = observed-to-expected lung-to-head ratio, GA = gestational age, IQR= interquartile range, POS = probability of survival, ECMO = extracorporeal membrane oxygenation. * X ^2^ test, # Mann–Whitney test, ° Fisher’s exact test.

**Table 6 children-10-00315-t006:** Intraoperatives related to length of stay.

	Overall (91)	>3 Weeks (54)	≤3 Weeks (37)	*p* Value
Side				0.18 *
Left, n (%)	79 (86.8%)	49 (90.7%)	30 (81%)	
Right, n (%)	12 (13.2%)	5 (9.3%)	7 (19%)	
Hernia Sac, n (%)	15 (16.4%)	7 (12.9%)	8 (21.6%)	0.27 *
Size				**<0.001 ***
A, n (%)	15 (16.5%)	6 (11.1%)	9 (24.3%)	
B, n (%)	38 (41.7%)	16 (29.6%)	22 (59.5%)	
C, n (%)	28 (30.8%)	23 (42.6%)	5 (13.5%)	
D, n (%)	10 (11%)	9 (16.7%)	1 (2.7%)	
Liver-up, n (%)	30 (32.9%)	19 (35.1%)	11 (29.7%)	0.59 *
Spleen-up, n (%)	57 (62.6%)	39 (72.2%)	18 (48.6%)	0.15 *
Patch, n (%)	29 (31.8%)	27 (50%)	2 (5.4%)	**<0.001 ***
Time for Surgery, minutes, median (IQR)	102 (75–130)	120 (101–160)	95 (81–120)	**0.001 #**

n = number, IQR = interquartile range, * X ^2^ test, # Mann–Whitney test.

**Table 7 children-10-00315-t007:** Postoperatives and therapeuticals related to length of stay.

	Overall (91)	>3 Weeks (54)	≤3 Weeks (37)	*p* Value
PH				**0.004 °**
None, n (%)	63 (69.2%)	32 (59.3%)	31 (83.7%)	
Mild, n (%)	10 (11%)	7 (12.9%)	3 (8.2%)	
Moderate, n (%)	7 (7.7%)	6 (11.1%)	1 (2.7%)	
Severe, n (%)	10 (11%)	9 (16.7%)	1 (2.7%)	
Unknown, n (%)	1 (1.1%)	0 (0%)	1 (2.7%)	
iNO_2_, n (%)	16 (10.9%)	15 (27.7%)	1 (2.7%)	**0.002 ***
Sildenafil, n (%)	3 (3.2%)	3 (5.5%)	0 (0%)	0.268 °
Inotropes, n (%)	64 (70.3%)	45 (83.3%)	19 (51.3%)	**0.002 ***
Dopamine, γ/kg/min, median (IQR)	8 (0–10)	10 (5–12)	6 (0–10)	**<0.001 #**
O_2_ at 30 days, n (%)	7 (7.6%)	7 (12.9%)	0 (0%)	**0.04 °**
Mechanical Ventilation, days, median (IQR)	9 (6–14)	12 (7–20)	6 (5–8)	**<0.001 #**

n = number, PH = pulmonary hypertension, iNO_2_ = inhalator nitric oxide, IQR = interquartile range, * X ^2^ test, # Mann–Whitney test, ° Fisher’s exact test.

## Data Availability

Not applicable.

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
