# Peer review of "Congenital Diaphragmatic Hernia: Perinatal Prognostic Factors and Short-Term Outcomes in a Single-Center Series"

_children, 2023, doi:10.3390/children10020315_

Round 1

Reviewer 1 Report

First, I would like to congratulate the authors for a well-conducted study. In this retrospective study, all newborns with CDH managed during the study period were included. Preoperative, intraoperative, and postoperative predictors of mortality and length of stay were assessed. Multivariable logistic regression was performed for the outcomes of death and length of 97 hospital stay 

The authors have demonstrated numerous factors predicting mortality and length of stay. However, on a multivariate regression model, the need for patch repair, and maximum dopamine dose were the only independent parameters associated with the length of stay.

The study has merit and will be of interest to our readers. I have a few comments:

Introduction: What did you hypothesize before conducting this study? Please write your hypothesis in 2-3 lines at the end of this section.

Methods: Well-written. No changes are needed.

Results: In the manuscript, this section is very disorganized. I would advise the authors to have subsections. Please segregate them into subsections: baseline characteristics, predictors of mortality, predictors of the length of stay, etc. 

Discussion: As in any retrospective study, there is incomplete/missing data. This is a major limitation of this study. The authors have mentioned this limitation in the last paragraph of the Discussion section.

Author Response

Please, see the file attached.

Reviewer 2 Report

Dear authors, you have a valuable study, which can be improved by adding some requirements. Good luck!

Author Response

Please, see the file attached.
